# Isotope systematics of subfossil, historical, and modern *Nautilus macromphalus* from New Caledonia

**Benjamin J. Linzmeier**[1]*, **Andrew D. Jacobson**[2], **Bradley B. Sageman**[2], **Matthew T. Hurtgen**[2], **Meagan E. Ankney**[3], **Andrew L. Masterson**[2], **Neil H. Landman**[4]

**1** Department of Earth Sciences, University of South Alabama, Mobile, Alabama, United States of America,
**2** Department of Earth and Planetary Sciences, Northwestern University, Evanston, Illinois, United States of America, **3** Wisconsin State Laboratory of Hygiene, University of Wisconsin–Madison, Madison, Wisconsin, United States of America, **4** Division of Paleontology, American Museum of Natural History, New York, New York, United States of America

* blinzmeier@southalabama.edu

**Data Availability Statement:** All stable isotope and elemental data are available in the Supplemental data file (SI 1) and are published in the EarthChem Library (https://doi.org/10.26022/IEDA/112707).

## Abstract

Cephalopod carbonate geochemistry underpins studies ranging from Phanerozoic, global-scale change to outcrop-scale paleoecological reconstructions. Interpreting these data hinges on assumed similarity to model organisms, such as *Nautilus*, and generalization from other molluscan biomineralization processes. Aquarium rearing and capture of wild *Nautilus* suggest shell carbonate precipitates quickly (35 μm/day) in oxygen isotope equilibrium with seawater. Other components of *Nautilus* shell chemistry are less well-studied but have potential to serve as proxies for paleobiology and paleoceanography. To calibrate the geochemical response of cephalopod $\delta^{15}N_{org}$, $\delta^{13}C_{org}$, $\delta^{13}C_{carb}$, $\delta^{18}O_{carb}$, and $\delta^{44/40}Ca_{carb}$ to modern anthropogenic environmental change, we analyzed modern, historical, and subfossil *Nautilus macromphalus* from New Caledonia. Samples span initial human habitation, colonialization, and industrial $p$CO₂ increase. This sampling strategy is advantageous because it avoids the shock response that can affect geochemical change in aquarium experiments. Given the range of living depths and more complex ecology of *Nautilus*, however, some anthropogenic signals, such as ocean acidification, may not have propagated to their living depths. Our data suggest some environmental changes are more easily preserved than others given variability in cephalopod average living depth. Calculation of the percent respired carbon incorporated into the shell using $\delta^{13}C_{org}$, $\delta^{13}C_{carb}$, and Suess-effect corrected $\delta^{13}C_{DIC}$ suggests an increase in the last 130 years that may have been caused by increasing carbon dioxide concentration or decreasing oxygen concentration at the depths these individuals inhabited. This pattern is consistent with increasing atmospheric CO₂ and/or eutrophication offshore of New Caledonia. We find that $\delta^{44/40}Ca$ remains stable across the last 130 years. The subfossil shell from a cenote may exhibit early $\delta^{44/40}Ca$ diagenesis. Questions remain about the proportion of dietary vs ambient seawater calcium incorporation into the *Nautilus* shell. Values of $\delta^{15}N$ do not indicate trophic level change in the last 130 years, and the subfossil shell may show diagenetic alteration of $\delta^{15}N$ toward lower values.

**Funding:** Ubben Program for Climate and Carbon Science at Northwestern University to BJL, ADJ, BBS, and MTH David and Lucile Packard Foundation (2007–31757) to ADJ The funders had no role in study design, data collection and analysis, decision to publish, or preparation of the manuscript.

**Competing interests:** The authors have declared that no competing interests exist.

Future work using historical collections of *Sepia* and *Spirula* may provide additional calibration of fossil cephalopod geochemistry.

## Introduction

The geochemistry of cephalopod mollusks has been used to quantify paleoclimate [1,2] and investigate paleobiology [3–9] throughout their long fossil record [10]. Modern *Nautilus* is a model organism for the isotope geochemistry of extinct cephalopods, including the ammonoids and nautiloids, because it is the only remaining externally shelled cephalopod (Fig 1). *Nautilus* are forereef scavengers that cross hundreds of meters of water depth within individual days while continuously growing shell. Extensive work has been done to quantify the systematics of light stable isotopes ($\delta^{13}C$ and $\delta^{18}O$) in *Nautilus* shell carbonate. Aquarium rearing experiments suggest *Nautilus* shells precipitate in $\delta^{18}O$ equilibrium with seawater [11,12]. Carbon isotope values ($\delta^{13}C_{shell}$) suggest both metabolic and dissolved inorganic carbon (DIC) sources [11] similar to that for other mollusks [13]. Analyses of $\delta^{15}N$ from shell organic matter, also known as conchiolin, show trophic change associated with hatching and maturity [14]. The carbon isotope value of conchiolin, like organic matter in other mollusk shells, likely reflects the $\delta^{13}C$ of respired carbon [13,15,16].

Other isotope systems, particularly metal isotopes, have received less investigation. The calcium isotope value ($\delta^{44/40}Ca$) of *Nautilus* shell has only recently been measured from a handful of modern shells [20,21] and may reflect changing *Nautilus* fractionation factor ($\Delta^{44/40}Ca_{shell-SW}$ [22], changing $\delta^{44/40}Ca$ value of seawater [23], or dietary change [20,21].

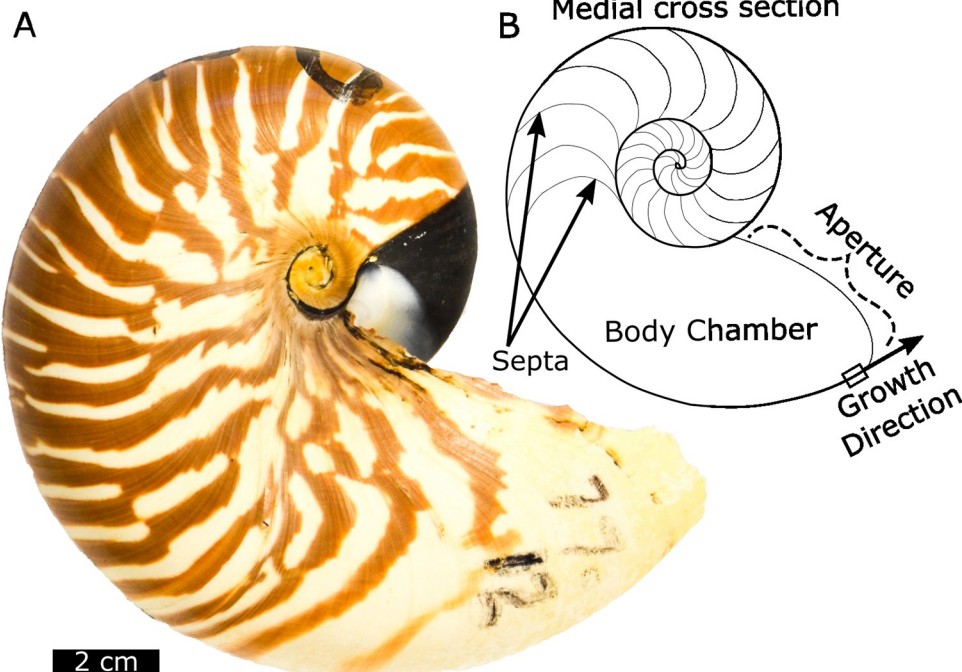

**Fig 1. Overview of *Nautilus m*. shell (ROM 48997) with annotation of major shell structures, and growth direction.** *Nautilus* shells grow in a spiral and septa are secreted to allow for the maintenance of neutral buoyancy during growth through the removal of liquid from the chambers [17]. Shells comprise two major structural forms of aragonite, the inner nacreous layer and outer spherulitic prismatic layer [18,19].

Proxies respond to anthropogenic forcings of environmental change. Some of these shifts are particularly dramatic and likely to impact *Nautilus*, including the rise of coastal 'dead zones' [24], overfishing [25], and acidification [26]. For most organisms, these environmental changes are unfolding on a generational timescale, so phenotypic plasticity and ecosystem feedbacks may allow for faster adaptation, potentially mitigating the worst possible effects [27–30]; however, it is difficult to assess deep-time analogs for change on timescales of 100's to 10,000's of years due to the time-averaged and punctuated nature of the fossil record [31,32]. Careful, high precision analyses of proxies across modern to historical samples provide an opportunity to calibrate the geochemical changes expected in fossil archives and provide insight into variations expected to be preserved within fossil assemblages during abrupt climate events.

Here, we use museum collections to test for hypothesized anthropogenic forcings (e.g. temperature change, ocean acidification, trophic change) and diagenetic vulnerability present in a suite of analyses (Sr/Ca, Mg/Ca, $\delta^{13}C_{org}$, $\delta^{15}N_{org}$, $\delta^{13}C_{carb}$, $\delta^{18}O_{carb}$, and $\delta^{44/40}Ca_{carb}$) of *N. macromphalus* from New Caledonia (Fig 2). Samples range in age from modern to subfossil and span indigenous settlement [33], European colonization, and industrial ocean acidification and warming (Fig 3). We hypothesize abrupt changes in ecologically sensitive proxies (e.g. $\delta^{18}O_{carb}$, $\delta^{15}N_{org}$) are stepwise due to ecological restructuring, while gradual changes are forced due to continuous industrial environmental effects. These data highlight the utility of museum collections and the subfossil record for better quantifying past environmental change to predict biological resiliency in the face of anthropogenic pressures.

## Materials and methods

### Ethics statement

All specimens of *N. macromphalus* (Mollusca: Cephalopoda) used in this study were collected for morphological study of *Nautilus* and deposited at various museums (American Museum of Natural History, Field Museum of Natural History, Royal Ontario Museum). The methods of

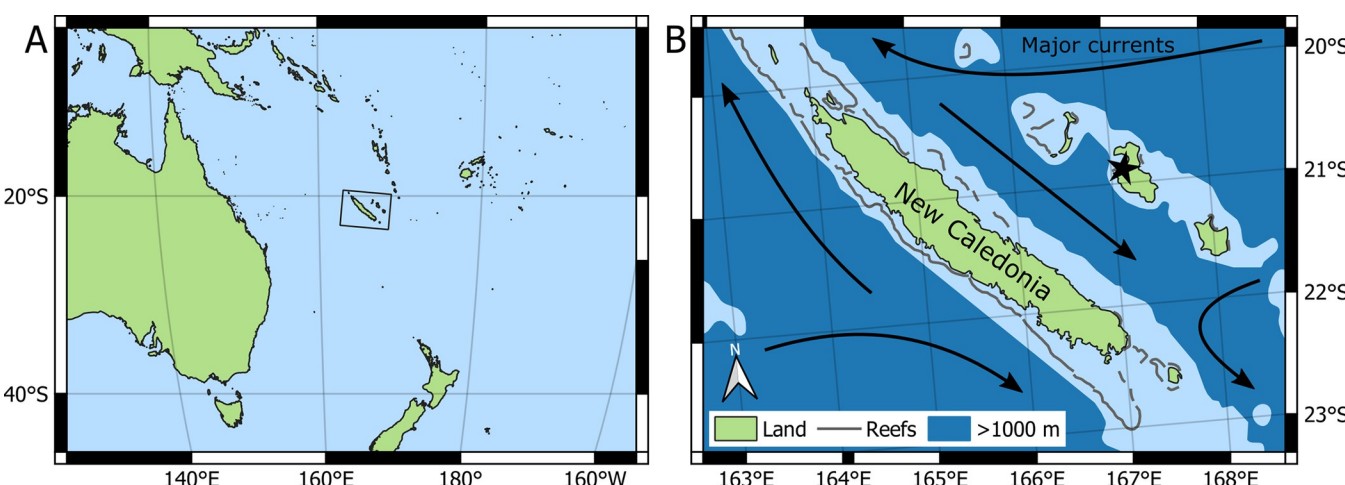

**Fig 2. Map showing the sample collection location and major currents surrounding New Caledonia.** A) World map showing the location of New Caledonia in the Southwestern Pacific Ocean. B) Local map showing New Caledonia and the surrounding islands with a star indicating the location of the cenote. Water shallower than 1000 m is in light blue and approximates the habitat of N. *macromphalus*. around New Caledonia. All cenote specimens are from the star location of Lifou, Loyalty Islands [34]. All other specimens were collected near the large island and are recent museum specimens. Reliable accounts of the live collection of *N. macromphalus* are mostly known from areas close to New Caledonia, and the species is thought to have a geographic range restricted to that region [35]. Local currents are overlain for reference [36].

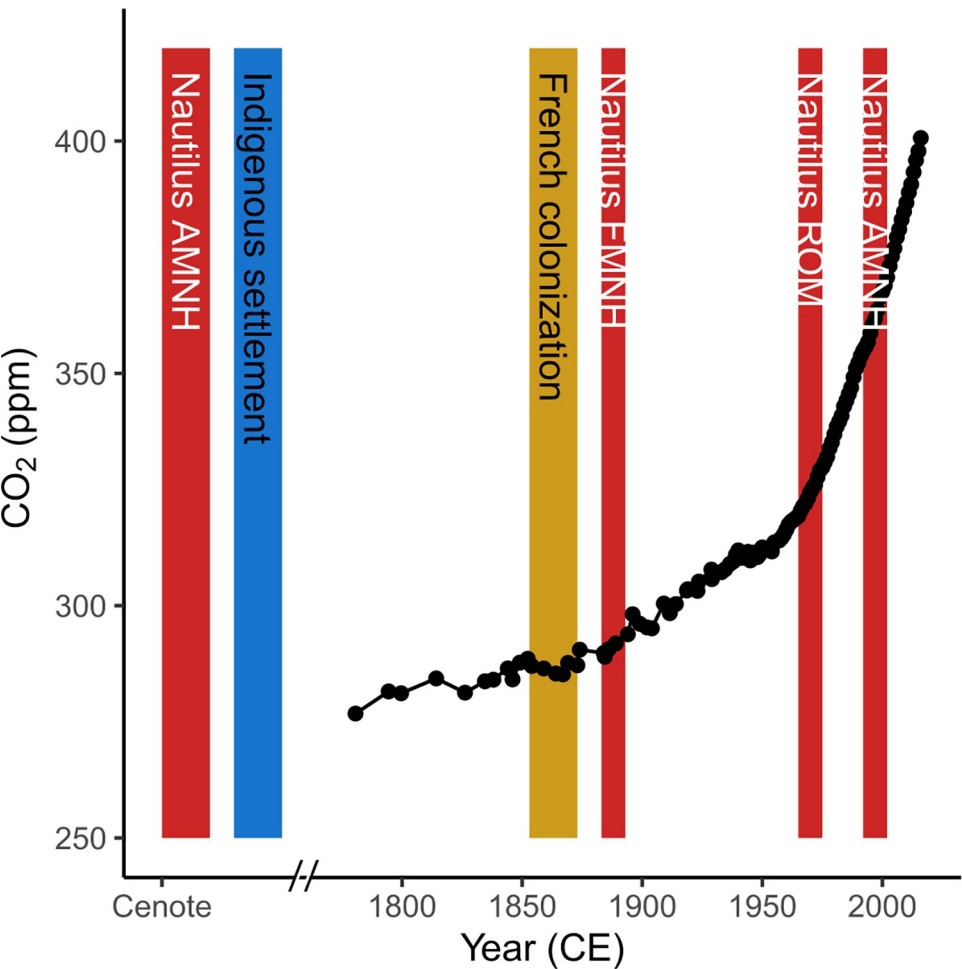

**Fig 3. Timeline for sample ages, pCO₂, and human settlement of New Caledonia.** Red vertical bars represent the age of specimens used in this study with the museum repositories noted (American Museum of Natural History—AMNH, Field Museum of Natural History—FMNH, and Royal Ontario Museum—ROM). Yellow vertical bar represents indigenous settlement as demonstrated by landscape change observed in lacustrine sediments ~3000 BCE [33]. French invasion occurred in the mid-1800's. The comparative record of pCO₂ change showing industrial increases is from Law Dome, East Antarctica [37].

capture and sacrifice of Nautilus before the 2000's specimens are unknown, but all specimens were collected before local protected status was granted in 2008. The two individuals collected in 2002 and 2004 were captured in a baited trap by Dr. Royal H. Mapes, sacrificed, and kept for a morphological study.

### *Nautilus macromphalus* samples

*N. macromphalus* is endemic to New Caledonia and the Loyalty Islands and can be distinguished by the prominent, open umbilicus with inward sloping walls ([38], Fig 1). Individual *Nautilus* likely grow to maturity over 2.5 to 6 years, although considerable uncertainty remains in their total lifespan [39]. The oldest samples used in this study are AMNH 112600 and AMNH 112595. These samples were recovered from a cenote on Lifou (Fig 2) and date to 6455 ±30 (AMNH 112600) and 6895 ±30 (AMNH 112595) years before present using radiocarbon [34]. Multiple fragments from near the aperture of this shell were powdered to provide adequate material for analyses. The second oldest sample is from the Field Museum of Natural

History in Chicago (FM 41) and was part of the original collection purchased from Wards Scientific to start the collection after the World's Columbian Exposition in 1893. A single 2 cm$^2$ chip from near the aperture was removed for analysis. Three samples from the Royal Ontario Museum (ROM 60177, 48997, and 49000) were collected in 1975, 1977, and 1978 respectively. Small chips of approximately 2 cm$^2$ were removed from the aperture. The most recent samples, housed at the American Museum of Natural History (AMNH 105621, AMNH 123397), were collected in 2002 and 2004, respectively. One sample, AMNH 105621, was previously studied for $\delta^{18}O_{carb}$ variability by secondary ion mass spectrometry [12]. For this sample, powder was drilled from a chip that was embedded in epoxy.

## Organic matter $\delta^{13}C$ and $\delta^{15}N$

Processing of carbonate to extract organic matter was done in the Sedimentary Geochemistry Laboratory at Northwestern University. Powdered shell was reacted with room temperature 1 N HCl overnight. Smaller amounts of recent shells with abundant organic matter were reacted with smaller volumes of HCl necessary for complete dissolution and then dried down to isolate the organic matter. The subfossil shells from the cenote required ~1–2 g of shell for the analyses presented here. Organic matter was collected from solution using ashed glass filters. Either organic matter or organic matter plus filters were loaded into tin capsules and combusted at 980 ˚C on a Costech 4010 Elemental Analyzer, coupled to a ThermoFinnigan Delta V Plus mass spectrometer via a ConFlo IV interface in the Stable Isotope Laboratory at Northwestern University. Long term precision of $\delta^{13}C$ and $\delta^{15}N$ analyses (±0.4‰ and ±0.6‰, $2\sigma_{SD}$) was assessed by comparison with isotope standards IU acetanilide ($\delta^{13}C$ = -29.52±0.02‰, $\delta^{15}N$ = +1.18‰) and IU urea ($\delta^{13}C$ = -8.02±0.05‰, $\delta^{15}N$ = +20.17±0.06‰) supplied by Indiana University [40]. This method of measurement does not distinguish between carbonate-bound organic matter (intracrystalline) and non-carbonate bound organic matter (intercrystalline) but produces average $\delta^{13}C$ and $\delta^{15}N$ values for all organic matter isolated from the *Nautilus* shell.

## Carbonate $\delta^{13}C$ and $\delta^{18}O$

Analyses of carbonate powder were conducted in the Stable Isotope Laboratory at Northwestern University using a Thermo Gasbench II coupled with continuous flow of He carrier gas to a Thermo Delta V IRMS. Values were standardized to the Vienna Pee Dee Belemnite (VPDB) scale using NBS-18 (National Bureau of Standards, $\delta^{18}O$ = -23.2 ‰, $\delta^{13}C$ = -5.01 ‰, VPDB) and an in-house carbonate standard, Carrara Lago Marble (CLM $\delta^{18}O$ = -3.66 ‰, $\delta^{13}C$ = 2.31 ‰, VPDB). The CLM standard has been calibrated alongside NBS-19 and IAEA-603. A two-point calibration using CLM and NBS-18 was employed for the analyses presented here. Analytical uncertainties ($2\sigma_{SD}$) for these analyses are ±0.2‰ for $\delta^{18}O_{carb}$ and ±0.1‰ for $\delta^{13}C_{carb}$.

### Elemental analysis

Approximately 50 mg of powdered shell was placed in acid-washed HDPE test tubes and dissolved in ~10 mL of 5 wt% $HNO_3$. The mixtures were agitated on a rocker table for ~24 hours, centrifuged, and then filtered through 0.45 μm syringe filters. Solutions were diluted with 5 wt % $HNO_3$ to ensure elemental concentrations were within calibration ranges of standards (assuming near-to-pure calcium carbonate). Measurements were done using a Thermo Scientific iCAP 6500 ICP-OES using Argon carrier gas and a Cetac U-6000AT+ Ultrasonic Nebulizer in the Aqueous Geochemistry Laboratory at Northwestern University. Repeated measurements of NIST SRM 1643f were used to both monitor instrument stability and confirm data quality for Ca, Mg, Mn, Na, and Sr. Three replicates of 3mL of solution were

averaged for standards, blanks, and samples. Instrument drift was monitored using a standard, sample, standard bracketing technique with no more than 15 samples analyzed between standards. Standard measurements suggest an instrumental precision of ±5% (RSD, relative standard deviation) for each element (Ca ~0.5 ppm, Mg ~0.1 ppb, Mn ~0.1 ppb, Na ~6 ppb, and Sr ~5 ppb).

## Carbonate δ$^{44/40}$Ca

Calcium isotope values were measured in the Radiogenic Isotope Laboratory at Northwestern University using an optimized $^{43}$Ca-$^{42}$Ca double spike method implemented on a Thermo-Fisher Triton Multi-Collector Thermal Ionization Mass Spectrometer (TIMS) [41]. Sample solutions containing 50 μg of Ca were weighed into acid-cleaned Teflon vials, spiked, and equilibrated. Solutions were dried down overnight at 90˚C. Residues were then dissolved in 0.5 mL of 1.55N HCl, and Ca was separated from other elements by passing solutions through Teflon columns filled with Bio-Rad AG MP-50 cation exchange resin. Solutions were dried down once more and subsequently treated with two drops of 35 wt% $H_2O_2$ to oxidize organic compounds and finally two drops of 16N $HNO_3$ to convert Ca to nitrate form. Samples were then dissolved in 0.4 μL of 8N $HNO_3$ and split into 4 beads. One bead was loaded onto a single degassed tantalum filament assembly, between ~0.5 mm wide parafilm dams, and subsequently dried at 3.5 amperes after adding 0.5 μL of 10 wt% $H_3PO_4$.

A stable, 20V $^{40}$Ca ion beam was obtained after a 0.5 hr warm-up sequence. Measurement of $^{40}$Ca/$^{42}$Ca, $^{43}$Ca/$^{42}$Ca, and $^{43}$Ca/$^{44}$Ca ratios was accomplished using a three-hop collector cup configuration. Sample (SMP) $^{44}$Ca/$^{40}$Ca ratios are reported in delta notation relative to OSIL seawater (SW), where δ$^{44/40}$Ca$_{SMP}$ = [($^{44}$Ca/$^{40}$Ca)$_{SMP}$ / ($^{44}$Ca/$^{40}$Ca)$_{SW}$− 1] x 1000. Analyses of the OSIL SW standard yielded a reproducibility of ±0.04‰ (2σ$_{SD}$) during the analytical sessions including these samples. Analyses of NIST 915b produced an average δ$^{44/40}$Ca value of -1.14‰ ± 0.07‰ (2σ$_{SD}$) during the duration of the study. All stable isotope and elemental data are available in the S1 Data and are published in the EarthChem Library (https://doi.org/10.26022/IEDA/112707).

## Difference estimates

The difference estimates between smaller subsets of data are based on a Monte Carlo approach. We add a random error term to each analysis assuming the instrumental precision describes two standard deviations of a normal distribution for expected variability around our measured values. We then calculate the difference between groups for this new dataset and repeat this operation 1,000,000 times. Error estimates are reported as the 5th to 95th percentiles of the differences between the groups from the Monte Carlo method.

## Results

### Carbonate trace elements

Recent samples show no discernible temporal trends in either Mg/Ca or Sr/Ca ratios and display similar variation to the subfossil specimens (Fig 4). Material from subfossil cenote specimens has higher Sr/Ca (3.0 mmol/mol) and lower Mg/Ca (0.5 mmol/mol) than most historical samples (Fig 4).

### Carbonate isotopes

The cenote specimens fall within the range of δ$^{18}$O and δ$^{13}$C of shell carbonate measured for modern specimens (Fig 5). No change in δ$^{44/40}$Ca is apparent in these samples less than 120

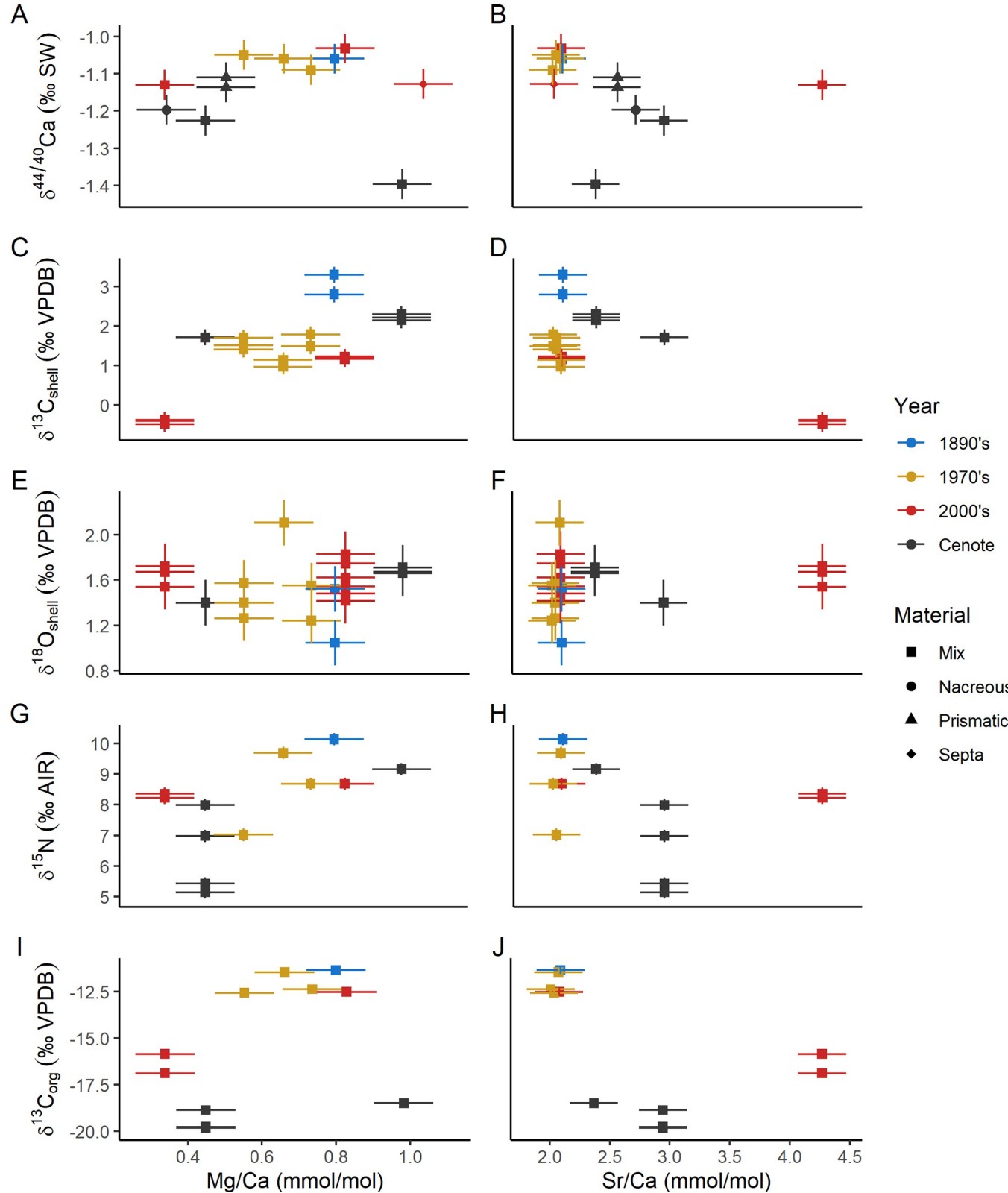

**Fig 4. Crossplots of Sr/Ca and Mg/Ca ratios vs isotope values measured on shell carbonate.** Error bars are 2 standard deviation instrumental precision for isotope values and the 5% RSD propagated to the elemental ratios. A) $\delta^{44/40}$Ca vs Mg/Ca, B) $\delta^{44/40}$Ca vs Sr/Ca, C) $\delta^{13}$C$_{shell}$ vs Mg/Ca, D) $\delta^{13}$C$_{shell}$ vs Sr/Ca, E) $\delta^{18}$O$_{shell}$ vs Mg/Ca, F) $\delta^{18}$O$_{shell}$ vs Mg/Ca, G) $\delta^{15}$N vs Mg/Ca, H) $\delta^{15}$N vs Sr/Ca, I) $\delta^{13}$C$_{org}$ vs Mg/Ca, and J) $\delta^{13}$C$_{org}$ vs Sr/Ca. The cenote and historical samples distinctly differ along Mg/Ca, Sr/Ca, $\delta^{15}$N, $\delta^{13}$C$_{org}$, and $\delta^{44/40}$Ca axes.

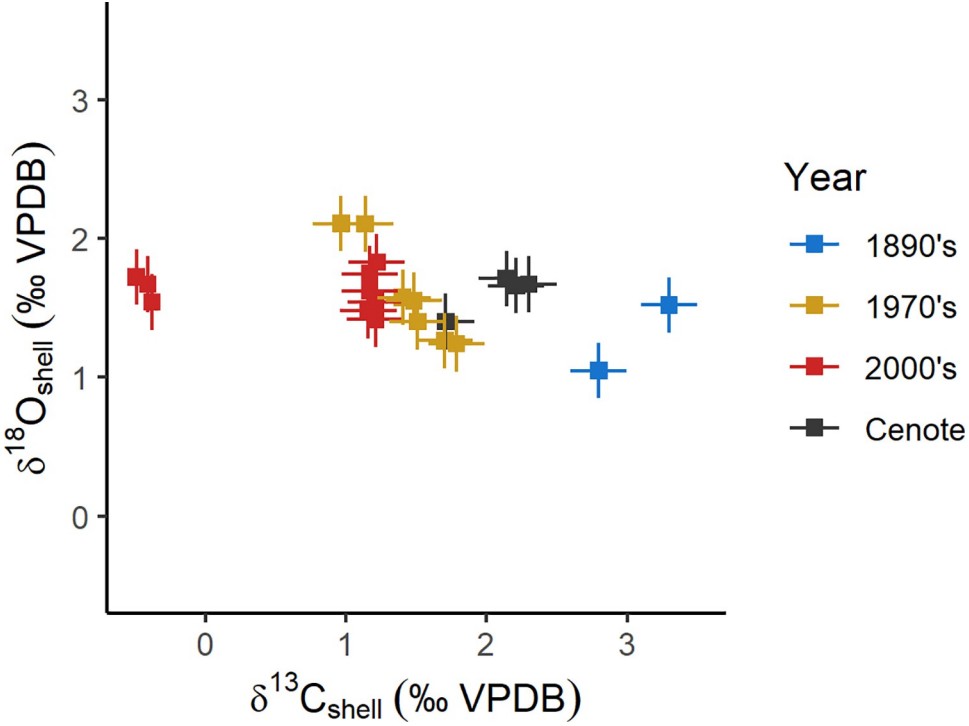

**Fig 5. Crossplot of carbon and oxygen isotope values measured on shell carbonate.** Error bars are 2 standard deviation instrumental precision.

years old; however, the subfossil cenote specimens are 0.12‰ lower (0.11 to 0.15‰) than modern samples (Fig 6A). For samples collected in the last 120 years, $\delta^{13}C_{shell}$ decreases by 2.09‰ (2.03 to 2.15‰) toward the present (Fig 6B), while $\delta^{18}O_{shell}$ increases by 0.32‰ (0.45–0.20‰) (Fig 6C).

## Organic matter isotopes

Samples spanning the last 130 years show no temporal trends in either $\delta^{13}C_{org}$ or $\delta^{15}N_{org}$ (Figs 6D, 6E and 7). Subfossil cenote specimens have lower $\delta^{13}C_{org}$ and $\delta^{15}N_{org}$ by 6.06‰ (6.25 to 5.87‰) and 1.74‰ (2.03 to 1.45‰), respectively. Recent samples also consistently have lower C/N (<3) compared to subfossil specimens (>4, Fig 7B).

## Discussion

### Diagenesis

**Elemental ratios.** Elemental ratios of biogenic carbonates are vulnerable to diagenetic alteration. Increases in Sr/Ca from 2 mmol/mol to 5 mmol/mol have been correlated with microstructural diagenetic alteration of fossil cephalopod shell and correspond to change in $^{87}Sr/^{86}Sr$, $\delta^{18}O_{carb}$, and $\delta^{13}C_{carb}$. The cenote samples do not appear to be outliers in either Sr/Ca or Mg/Ca compared to modern samples, however. This suggests that diagenetic alteration of Sr/Ca and Mg/Ca is not detectable in the cenote samples.

**Organic matter isotopes.** Although organic matter within mollusk shells has been reported for multiple fossil specimens, pristine preservation is rare [42–45]. During dissolution of these shells for isolation of organic matter, it is apparent that cenote specimens have

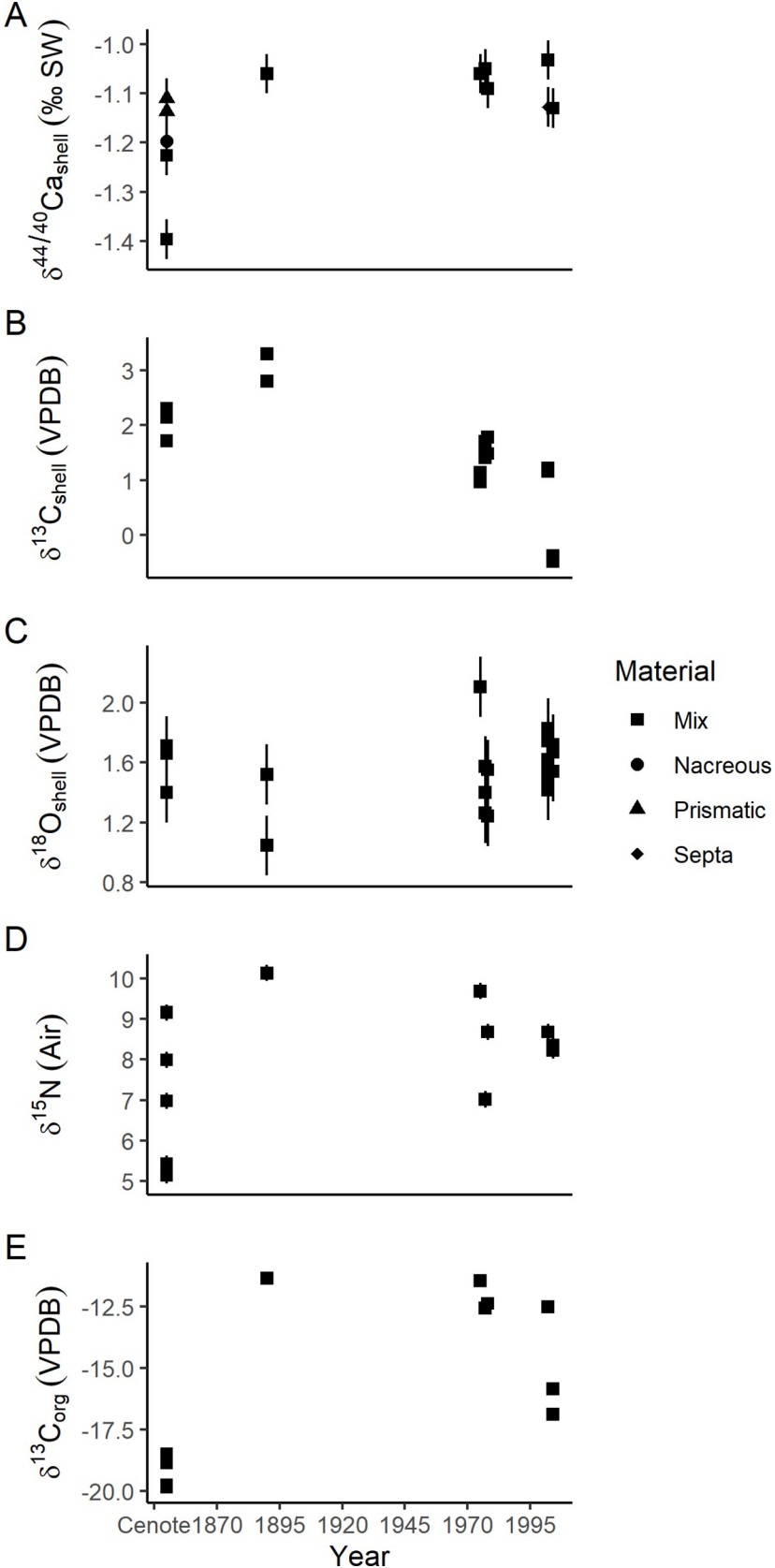

**Fig 6. Stable isotope values through time.** Error bars are 2 standard deviation instrumental precision. A) Shell $\delta^{44/40}$Ca through time. B) The $\delta^{13}$C of shell carbonate through time. C) The $\delta^{18}$O of shell carbonate through time. D) The $\delta^{15}$N of shell organic matter though time. E) The $\delta^{13}$C of shell organic matter through time.

considerably less organic matter than modern specimens, but the effects on the isotope composition of residual organic matter is important to explore. Degradation of organic matter within shells generally removes more labile organic matter (e.g. amino acids, glycoproteins, sugars) and preserves refractory organic matter, although the exact effect on $\delta^{13}C_{org}$ and $\delta^{15}N_{org}$ values is unknown. Two sources of organic matter in the shell, carbonate-bound organic matter (intracrystalline) and non-carbonate bound organic matter (intercrystalline), likely behave differently during diagenesis due to the differing volumes and connectedness of the organic matter for microbial degradation. Some authors have noted much lower $\delta^{15}$N (1– 4‰) in significantly older fossil nautiloids (*Cymatoceras sakalavus*, Albian ~113 to 100 Ma) and have suggested these low $\delta^{15}$N values reflect diagenetic alteration [14]. Sedimentary organic matter degradation drives $\delta^{15}$N toward lower values while C/N increases [46]. We see a similar pattern between the recent and subfossil specimens (Fig 7B) and therefore suggest that the $\delta^{15}$N of the cenote specimens is altered and does not reflect a primary trophic level signature. The $\delta^{13}C_{org}$ values in these samples are also lower than the more recent specimens (Fig 7A). Diagenetic alteration of $\delta^{13}C_{org}$ in mollusks also likely drives isotope values lower due to a loss of the hydrophilic fraction of organic matter and potentially transformation to hydrophobic organic matter. The transformation involves preferentially losing $^{13}$C by not reincorporating it in hydrophobic organic matter.

**Carbonate isotopes.** Diagenetic alteration of light stable isotopes is of particular interest for constraining the fidelity of paleoclimatological records [47–49]. The subfossil specimens from the cenote show signs that shell nacre recrystallized [34], and higher Sr/Ca in these shells is consistent with addition of either strontianite [47] or authigenic aragonite (Fig 4B). The lack of clear $\delta^{13}$C or $\delta^{18}$O outliers is notable and suggests that any recrystallization did not incorporate large volumes of additional diagenetic carbonate. Shell $\delta^{44/40}$Ca may have been altered toward lower values, as has been observed in Cretaceous mollusks from Antarctica [22], where

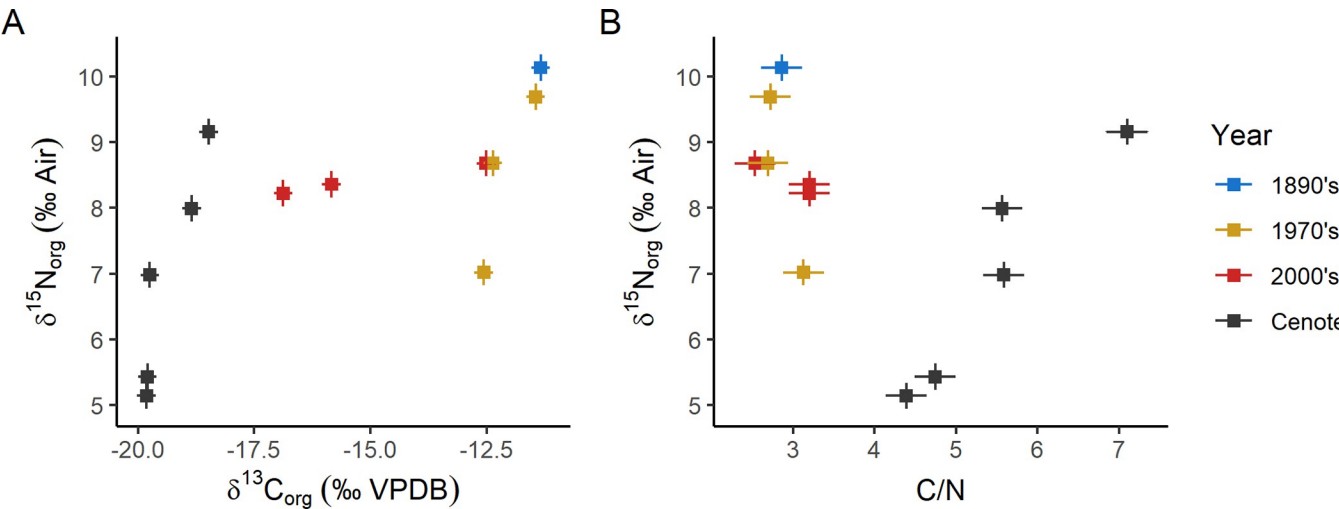

**Fig 7. Isotope values of organic matter and the C/N ratio of the organic matter.** Error bars are 2 standard deviation instrumental precision. A) $\delta^{13}C_{org}$ vs $\delta^{15}N_{org}$, B) C/N vs $\delta^{15}N_{org}$.

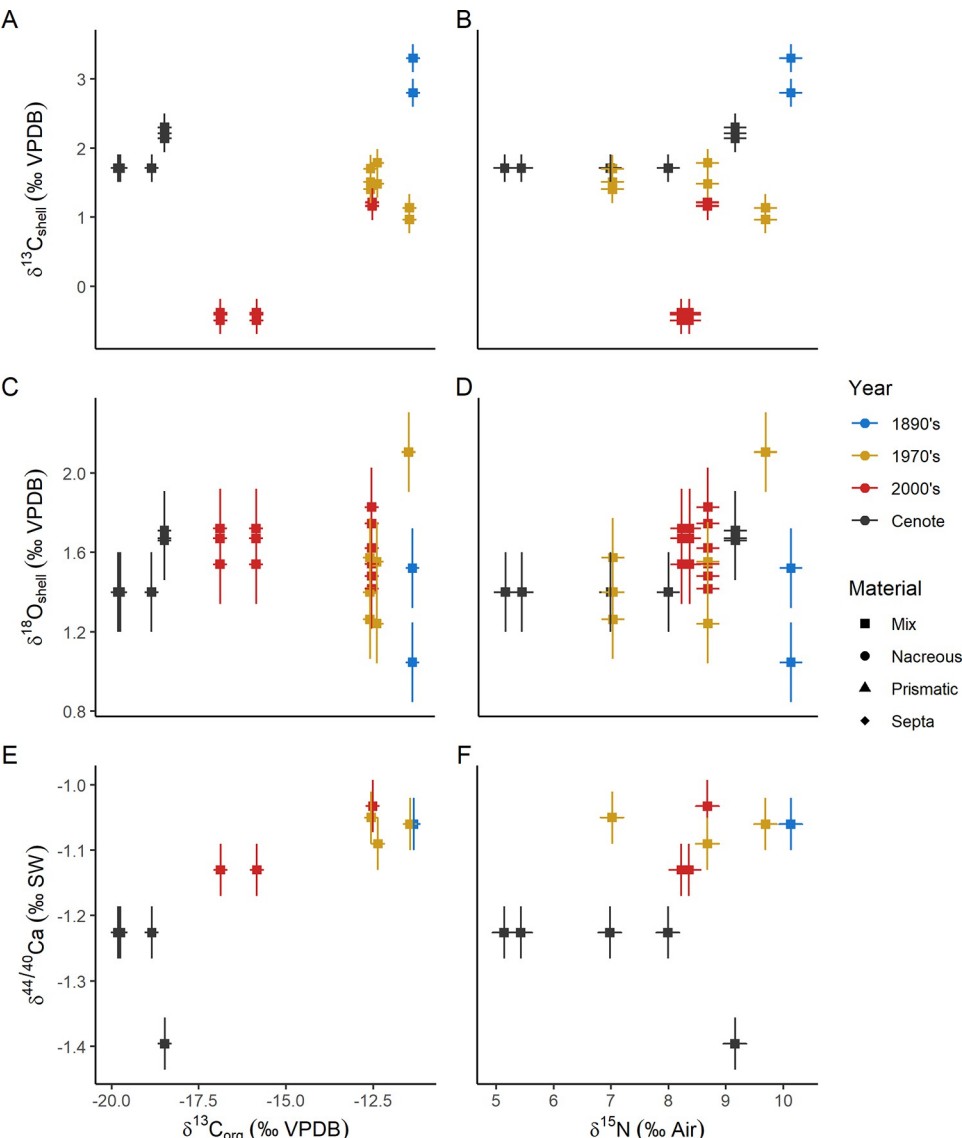

**Fig 8. Crossplots of isotope values measured in shell organic matter vs shell carbonate.** Error bars are 2 standard deviation instrumental precision. A) $\delta^{13}C_{org}$ vs $\delta^{13}C_{shell}$, B) $\delta^{15}N$ vs $\delta^{13}C_{shell}$, C) $\delta^{13}C_{org}$ vs $\delta^{18}O_{shell}$, D) $\delta^{15}N$ vs $\delta^{18}O_{shell}$, E) $\delta^{13}C_{org}$ vs $\delta^{44/40}Ca$, and F) $\delta^{15}N$ vs $\delta^{44/40}Ca$.

trends toward higher Sr/Ca (5 up to 12 mmol/mol) correlate with lower $\delta^{44/40}Ca$ values (from -1.5 to -2‰). This could point to the addition of small domains of authigenic aragonite or strontianite, but the variation observed in the Sr/Ca in the most recent *N. macromphalus* (Fig 4) suggests that the pattern may not be consistent with diagenesis.

If any diagenesis occurred, it happened at low temperatures (~28°C), ambient pressures, and within the last 7000 years [34] given the location of the cenote and previously published radiocarbon ages. Carbonate recrystallization/precipitation was likely tied to microbial degradation of organic matter causing variable carbonate saturation state within or near shells [50,51]. Future work contrasting these diagenetic pathways with higher temperature hydrothermal experiments [52,53] may help provide a more detailed 'taphonomic chronometer' for use in deep time [54].

## Isotope and elemental interpretation

**Elemental ratios.**    The elemental ratios of *Nautilus* shells potentially reflect taxonomic or geographic differences rather than temperatures, as has been observed in bivalve mollusks. We observe a wide range of Sr/Ca and Mg/Ca across these samples with no clear trend towards modern samples. This variation within modern specimens suggests that more work on Mg/Ca and Sr/Ca variation within *Nautilus* is needed to understand the limits of intraspecific variation and potential thresholds for identification of diagenetic alteration.

**Organic matter isotopes.**    Modern and historical shells suggest slight variation in prey preference or availability for the individuals measured (Fig 7A). In other work on *Nautilus*, $\delta^{15}N_{org}$ changes with hatching and early growth and variation of ~1–2‰ within and between individuals has been observed [14]. Similar individual–to–individual and within individual variability may be linked to differing prey choice [55,56] and is likely to be present in *Nautilus* given neurological complexity [57]. There are no strong trends through time in the recent $\delta^{13}C_{org}$ or $\delta^{15}N_{org}$ (Figs 6D, 6E and 7A) that would suggest major trophic changes for these *N. macromphalus* in the last ~120 years. Future work using compound specific $\delta^{15}N_{org}$ analysis of amino acids within samples like these could support trophic level stability.

**Calcium isotopes.**    Previous studies have proposed that $\delta^{44/40}Ca$ can indicate trophic level for vertebrates in modern and ancient terrestrial and marine ecosystems [58,59]. Ingestion of hard tissue material and subsequent use of calcium from that source in biomineralization is likely necessary to cause a $\delta^{44/40}Ca$ trophic level effect [60]. It is possible that the difference measured between the oldest *N. macromphalus* and the more recent samples (Figs 6A and 8F) reflects changing composition of scavenged prey, including crustaceans and fish [61], perhaps due to ecosystem changes coupled to fishing pressures [62,63]. Shifting dietary composition causing different $\delta^{44/40}Ca$ is an unlikely explanation, however, because no strong correlation with either $\delta^{15}N_{org}$ or $\delta^{13}C_{org}$ is observed, and moreover, recent individuals show measurable differences in both isotope systems indicative of differing diets (Fig 8F [55]). In addition, comparing $\delta^{44/40}Ca$ to average $\delta^{15}N$ and trophic level across other modern *Nautilus*, *Sepia*, and *Spirula* suggests a lack of trophic response in cephalopods [21].

The calcium isotope composition of mollusk material may vary with the calcium isotope composition of seawater ($\delta^{44/40}Ca_{SW}$) or changes in the fractionation between seawater and precipitated shell ($\Delta^{44/40}Ca_{SW-Shell}$) [22]. Bivalve mollusks derive ~80% or more of their shell calcium from seawater sources [64] and can make use of only water sources for 100% of calcium in shell precipitation [65]. While shifts in local-scale Ca cycling dynamics are possible [23], changes in the isotopic composition of the global ocean unlikely explain the observed variations, given the long residence time of Ca in seawater. Evidence from the geologic record, however, suggests that $\Delta^{44/40}Ca_{SW-Shell}$ is sensitive to seawater carbonate state. Culturing experiments show that the pH of extrapallial fluid from *Arctica islandica* bivalves is at least partially regulated during shell formation, but can vary with ambient pH [66]. If *Nautilus* similarly regulates the pH and potentially the saturation state of extrapallial fluid during shell precipitation, then variation in $\Delta^{44/40}Ca_{SW-Shell}$ would be driven by mass-dependent isotope effects during transport into the extrapallial fluid rather than with varying saturation state and precipitation rate at the site of precipitation, as has been observed in inorganic systems. Temperature may affect $\Delta^{44/40}Ca_{SW-Shell}$ in bivalves [67], although temperature covaries with carbonate saturation state, which can be modulated by the ratio of photosynthesis to respiration [68,69], and in general, the temperature sensitivity of $\Delta^{44/40}Ca_{SW-Shell}$ for other carbonate producers is low [20,70–73].

Seawater likely constitutes the largest source of calcium for the precipitation of *N. macromphalus* shell [64,65], but complex precipitation and resorption of intermediate carbonate

reservoirs could influence shell $\delta^{44/40}$Ca [74]. Uroliths represent one proposed intermediary carbonate structure between seawater and shell precipitation [75]. These structures are amorphous calcium phosphate and have higher Sr contents than the shell wall [76]. The calcium fractionation factor for this material is unknown, but may be close to -1‰, similar to peloidal calcium phosphate [77,78].

Fixed $\Delta^{44/40}$Ca$_{\text{SW-Shell}}$ in cephalopods (mostly for belemnites) is typically assumed in the reconstruction of $\delta^{44/40}$Ca$_{\text{SW}}$ change through time [21,79–81]. Recent evidence suggests that $\Delta^{44/40}$Ca$_{\text{SW-Shell}}$ for other types of carbonate producers varies with seawater saturation state [22,82,83] in a way consistent with theoretical models for inorganic calcite formation [84,85]. Any variation of cephalopod $\Delta^{44/40}$Ca$_{\text{SW-Shell}}$ may reflect additional controls, such as changes in the transport of Ca into the extrapallial fluid (EPF). One possibility is that proportions of Ca transported by selective intracellular calcium channels, non-selective intercellular pathways, or active enzymatic transport (Ca$^{2+}$-ATPase and carbonic anhydrase) can change. However, under most conditions, selective intracellular calcium channels are thought to dominate transportation of Ca into the EPF through high diffusive fluxes that are selective for Ca [86]. Another possibility is that any Ca isotope fractionation imparted by one or more of these transport mechanisms varies with seawater saturation state. Future well-controlled aquarium rearing experiments will be necessary to determine if $\Delta^{44/40}$Ca$_{\text{SW-Shell}}$ changes with carbonate saturation state or another environmental condition.

For anthropogenic ocean acidification to change $\Delta^{44/40}$Ca of these *Nautilus*, higher $CO_2$ concentrations must propagate to their living depths, which in turn must remain relatively stable throughout the study interval. The observed change in $\delta^{13}$C of shell carbonate (Fig 6B) suggests propagation of anthropogenic carbon into the habitat of the *Nautilus* but does not guarantee a change in the concentration of $CO_2$ necessary to cause acidification. Boron isotope measurements from shallow water (7–8 m) coral suggests a decrease of ~0.15 pH near New Caledonia since the 1890's, but also indicate similar variations on decadal timescales [87]. Deeper waters, like those that the *Nautilus* inhabit, are generally expected to acidify more slowly, although organic matter export can sometimes enhance acidification at depth depending on the interaction between deep water ventilation, organic matter export, and general ocean circulation patterns [88,89]. Because the oceans are still in the early stages of acidification, it is likely that the amount of pH change at these depths has not been sufficient to cause detectable alteration in the $\delta^{44/40}$Ca of the studied *Nautilus* specimens.

**Respired $CO_2$.** The proportion of respired carbon incorporated into the shells of mollusks depends on changes in ambient $O_2$:$CO_2$ in surrounding waters [13,90] and the metabolic rate [91]. Use of the proxy assumes that all carbon used for the precipitation of biogenic carbonates derives from either seawater dissolved inorganic carbon (DIC) or respired $CO_2$ derived from food. The $\delta^{13}$C metabolic rate proxy has been explored using aquarium-reared cephalopods, including *Sepia pharaonis* [92,93]. The proportion of respired carbon contributing to carbonate precipitation can be calculated following a simple relationship outlined by [13,90]:

$$C_{metabolic} = \frac{\delta^{13}C_{shell} - \varepsilon_{aragonite} - \delta^{13}C_{DIC}}{\delta^{13}C_{metabolic} - \delta^{13}C_{DIC}} \tag{1}$$

Where $\delta^{13}$C$_{\text{shell}}$ refers to the measured value for carbonate shell material. We assume $\delta^{13}$C$_{\text{metabolic}}$ is equivalent to $\delta^{13}$C$_{\text{org}}$ measured in the shell organic matter—given similar findings for other mollusks [16]. The value of $\varepsilon_{\text{aragonite}}$ is assumed to be 2.7‰, which is the value for inorganic aragonite [94]. To constrain $\delta^{13}$C$_{\text{DIC}}$, we use published estimates of $\delta^{13}$C$_{\text{DIC}}$ for preindustrial (1.2‰, VPDB) and recent (1.0‰, VPDB) water near New Caledonia [95,96] at the average *N. macromphalus* living depth of ~400 m [97]. This depth is chosen even though shell

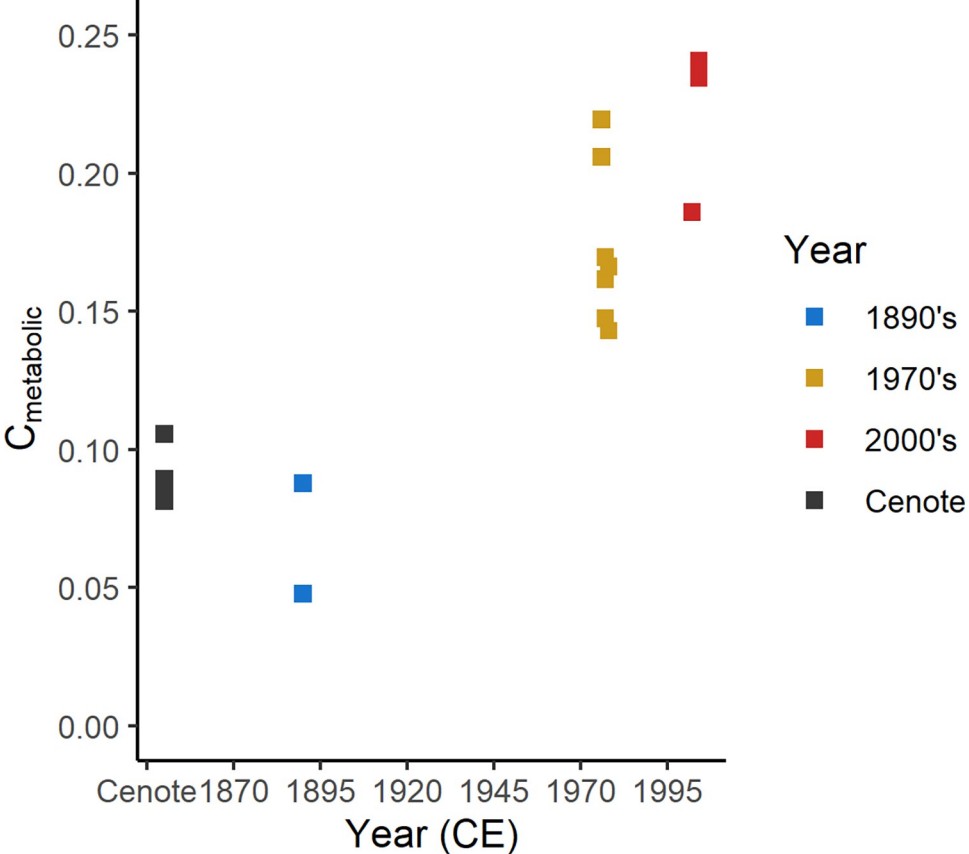

**Fig 9. Crossplot of the calculated metabolic carbon contribution vs time.** Note the increase in the amount of metabolic carbon incorporated into the shell since the 1970's.

grows across a range of at least 400 m water depth [12] due to time averaging of bulk samples like those used in this study [98]. We assume no ontogenetic change in $\delta^{13}C_{carb}$ biases our results. All samples were from near at the aperture of the body chamber of large shells, and most change in $\delta^{13}C_{carb}$ has been observed in the shell near to hatching rather than later in ontogeny, as is observed in bivalves.

Calculations using these metrics show increasing amounts of respired carbon are being incorporated into the shells of *N. macromphalus* during the last 130 years (Fig 9). The observed increase (from ~8 to ~22%) likely reflects changing metabolic rates caused by transgenerational acclimation to ocean acidification, as has been seen in *Mya arenaria* bivalves [91]. The increase is much less than the observed metabolic carbon increase (27% to 61%) in the experimentally reared bivalves over what is likely to be a much larger pH range (8.1 to 7.7 pH) [91]. Other experimentation with *Arctica islandica* bivalves has suggested that ontogenetic variation in the amount of respired carbon incorporated into shells is minimal [99]. Because the *Nautilus* live relatively deep, $CO_2$ incorporation from the atmosphere is likely to be lower than that experienced by shallower organisms. The decrease of pH at the living depths of these *N. macromphalus* may be facilitated by dampened deep water oxygenation related to continued organic matter export [88]. The observed increase in respired carbon could also be modulated by changes in the ratio of dissolved oxygen to carbon dioxide through expanding oxygen minimum zones offshore of New Caledonia due to increased agricultural intensity and nutrient-rich runoff [100,101], dramatic changes to vegetation [102], increasing [$CO_2$] due to

anthropogenic injection into the atmosphere-ocean system [103], pressure from mining runoff [104], or changing preferred habitat depths due to changing ecosystem composition from fishing other organisms (Fig 3). It is also unlikely that the observed increase in metabolic carbon is caused by ontogenetic change (e.g., because shells represent mature *Nautilus* and most $\delta^{13}C_{carb}$ change occurs very early in ontogeny.

## Future work

One of the primary challenges to clearly calibrating geochemical proxies in biogenic carbonates is the 'shock' of changing environmental conditions during the experimental treatment of individuals [105,106]. Although the magnitude of anthropogenic and natural environmental change can be large, both are slow relative the generational timescale of organisms. This means that the best calibrations for proxies preserved in biogenic carbonates may be found in historical or spatially distributed samples that inhabited conditions across several generations [29,105]. Studies in modern settings may benefit from cross-calibration with low-cost real-time environmental logging [107]. Focus on mollusk shell is of particular merit due to wide ranging methods for diagenetic assessment [47,48,53,108,109] and preservation of shell organic matter [43,44]. Future work on these isotope systems in cephalopods and other mollusks should leverage museum collections spanning collection time and geographic range to avoid confounding factors of shock and to allow for adaptive responses. Further work on internally shelled mollusks (*Sepia* and *Spirula*) may be informative for interrogating belemnite archives given newfound microstructural complexity [110,111].

## Conclusions

Geochemical analyses of subfossil, historical, and modern *N. macromphalus* show the importance of calibrating geochemical signals across generations of organisms. The low temperature, potential diagenetic alteration also points to the importance of robust diagenetic assessment of samples from different environments. Our data suggest that there is intergenerational increase in *N. macromphalus* metabolism near New Caledonia, which is likely caused by ocean acidification. Future work using historical and modern marine mollusk shell material will be important for calibrating geochemical proxies across events that can cause both selection and/or phenotypic plasticity response.

## Supporting information

**S1 Data. Spreadsheet with all analytical data for all samples.** Each tab contains one group of analyses.
(XLSX)

## Acknowledgments

We would like to thank Jochen Gerber (Field Museum) and Bushra Hussaini (AMNH) for assistance in accessing samples. We also thank three anonymous reviewers for their constructive comments to improve this manuscript.

## Author Contributions

**Conceptualization:** Benjamin J. Linzmeier, Andrew D. Jacobson, Bradley B. Sageman, Matthew T. Hurtgen.

**Data curation:** Benjamin J. Linzmeier.

**Formal analysis:** Benjamin J. Linzmeier.

**Funding acquisition:** Andrew D. Jacobson, Bradley B. Sageman, Matthew T. Hurtgen.

**Investigation:** Benjamin J. Linzmeier.

**Methodology:** Andrew D. Jacobson, Meagan E. Ankney, Andrew L. Masterson.

**Resources:** Andrew D. Jacobson, Neil H. Landman.

**Supervision:** Andrew D. Jacobson, Bradley B. Sageman, Matthew T. Hurtgen.

**Visualization:** Benjamin J. Linzmeier.

**Writing – original draft:** Benjamin J. Linzmeier.

**Writing – review & editing:** Benjamin J. Linzmeier, Andrew D. Jacobson, Bradley B. Sageman, Matthew T. Hurtgen, Meagan E. Ankney, Andrew L. Masterson, Neil H. Landman.

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
