## [Decision Letter · Decision Letter 0]

5 Jan 2022

PONE-D-21-31441Isotope systematics of subfossil, historical, and modern Nautilus macromphalus from New CaledoniaPLOS ONE

Dear Dr. Linzmeier,

Thank you for submitting your manuscript to PLOS ONE. After careful consideration, we feel that it has merit but does not fully meet PLOS ONE’s publication criteria as it currently stands. Therefore, we invite you to submit a revised version of the manuscript that addresses the points raised during the review process.

The reviewers highlight the value of the presented dataset of natural variability as recorded in Nautilus shells which complements previous culture experiments. They however raised several key areas where improvement is required and several issues that need to be further discussed or clarified, pertaining for instance to the methodology and presentation of the data but not only. The manuscript would also benefit from a clarification of the goal of the study, and from clearer outlining of the hypotheses. Therefore we encourage you to revise your manuscript taking into account these recommandations.

We look forward to receiving your revised manuscript.

Kind regards,

Emmanuelle Puceat

Academic Editor

PLOS ONE

Journal Requirements:

Ubben Program for Climate and Carbon Science at Northwestern University to BJL, ADJ, BBS, and MTH

David and Lucile Packard Foundation (2007–31757) to ADJ

NO

4. We note that Figure 2 in your submission contain [map/satellite] images which may be copyrighted. All PLOS content is published under the Creative Commons Attribution License (CC BY 4.0), which means that the manuscript, images, and Supporting Information files will be freely available online, and any third party is permitted to access, download, copy, distribute, and use these materials in any way, even commercially, with proper attribution. For these reasons, we cannot publish previously copyrighted maps or satellite images created using proprietary data, such as Google software (Google Maps, Street View, and Earth). For more information, see our copyright guidelines: http://journals.plos.org/plosone/s/licenses-and-copyright.

Reviewers' comments:

Reviewer's Responses to Questions

**Comments to the Author**

1. Is the manuscript technically sound, and do the data support the conclusions?

Reviewer #1: Yes

Reviewer #2: No

Reviewer #3: Yes

2. Has the statistical analysis been performed appropriately and rigorously? 

Reviewer #1: No

Reviewer #2: Yes

Reviewer #3: N/A

3. Have the authors made all data underlying the findings in their manuscript fully available?

Reviewer #1: Yes

Reviewer #2: Yes

Reviewer #3: Yes

4. Is the manuscript presented in an intelligible fashion and written in standard English?

Reviewer #1: Yes

Reviewer #2: Yes

Reviewer #3: Yes

5. Review Comments to the Author

Reviewer #1: Dear authors,

In their manuscript, Linzmeier et al. present a multi-proxy dataset measured in modern and sub-fossil Nautilus macromphalus shell carbonate. The goal of the study is to investigate the effect of several environmental parameters on the shell composition of N. macromphalus to provide a baseline for studies on historical and fossil specimens of this and related species aiming to reconstruct environmental change. The authors present a valuable dataset of natural variability as recorded in Nautilus shells which complements previous culture experiments. The decision to work on shells from historical collection is interesting, as it avoids certain complications of culture experiments (most notably the shock to the animals due to changing aquarium conditions) as well as the ethical considerations of experimenting on live cephalopods. Overall, I believe this manuscript merits publication barring some moderate revisions, mostly pertaining to the methodology and the presentation of the data, which I will highlight in my line-by-line comments below:

Line 42: “isotope systematics of light stable isotopes” rephrase to “systematics of light stable isotopes”

Line 92: The authors should provide the uncertainty on the radiocarbon date of the cenote specimen in the text

Lines 104-115: Recently, there has been some discussion in the literature about the pre-treatment methods required to distinguish between the contributions of skeletal carbonate-bound organic matter and non-carbonate bound organic matter phases in d13C and d15N analyses of biocarbonates (see Ren et al., 2009; Straub et al., 2013; Gillikin et al., 2017; de Winter et al., 2021). It seems like the pre-treatment method used in this study (simple ashing and acid treatment) does not allow this distinction to be made, while the difference between the sources of organic matter are likely important for the diagenesis discussion (section 4.1.1). I suggest that the authors add some discussion of their methodology in comparison to the state of the art and discuss how they think the preservation of different sources of organic matter may impact d13C and d15N measured in (sub-)fossil Nautilus specimens. Perhaps they could also give some recommendations for sample treatment that would allow future authors to capture the original isotopic composition of organic matter in Nautilus shells.

Line 108: “with less HCl” please provide the amount and concentration of the acid such that the experiments can be reproduced.

Line 120-121: Only one international standard was used to correct the carbonate isotope composition. This procedure does not allow testing of potential scale compression by the mass spectrometer, which influences the accuracy of the measurements without affecting the reproducibility (precision) measured on the independent standard. If the authors have data on (an) additional independent standard(s) measured in the same routine, I suggest they add this data to show that this scale compression effect is not an issue.

Line 131: “Standard measurements suggest an instrumental precision of ±5% (RSD).” Please mention which elements were measured here. It is not clear for which element(s) this precision is reported. It is highly unlikely that all measured elements have the same measurement precision. The authors should add more detailed statistics of accuracy and precision for all measured elements here in the main text.

Line 157-158: The modern shell (2000 CE) also has higher Sr/Ca and lower Mg/Ca values than the subfossil ones, but this is not really discussed in the manuscript. On the contrary, the authors make the case that the higher Sr/Ca and lower Mg/Ca values in the cenote sample are a consequence of diagenetic alteration (see section 4.1.2), but this does not explain why the modern shell also has higher Sr/Ca and lower Mg/Ca. I therefore think the paper could benefit from a more detailed discussion of the factors that may influence trace element ratios in Nautilus shells.

Line 162: “~0.12‰ lower” An assessment of the difference in isotopic value like this requires statistics. The authors should report the exact difference and the propagated error on this difference to demonstrate that it is statistically significant.

Line 167-168: “~7‰ and ~3‰, respectively” See comment above: these comparisons should be tested statistically (e.g. using a Student’s t-test) to show the reader whether the difference is significant.

Line 177: “Some have noted much lower δ15N in much older fossil nautiloids” Rephrase to “Some authors have noted…”. Here I think a more thorough discussion of the different sources of carbon and nitrogen in the shell (e.g. carbonate-bound and non-carbonate-bound) and how they influence the d13C and d15N values is warranted. Especially since the comparison is made with very old fossil specimens.

Line 183-185: Here, several reasons are given for the difference in d13C between the subfossil and modern specimens. However, contrary to the d15N value, the possibility of diagenesis is not discussed as far as I can see. Can the authors be certain that diagenetic processes did not affect d13C but did affect d15N? If so, they should add some evidence to support this assumption.

Line 193-196: “Shell d44/44Ca… …or strontianite.” I am not convinced that the authors demonstrate the correlation between calcium isotope value and Sr/Ca ratios which hints at diagenesis. The fact that the Sr/Ca values in the modern shells are highest actually contradicts this, and this is not discussed. The authors should discuss these trends in more detail if they want to convincingly demonstrate the diagenetic effect on the calcium isotopes.

Section 4.2.2 is one of the most interesting parts of the manuscript, and the discussion here should form a nice benchmark for interpreting future cephalopod calcium isotope measurements.

Lines 237-239: “Culturing experiments show that the pH of extrapallial fluid from Arctica islandica bivalves is at least partially regulated during shell formation, but can vary with ambient pH (Liu et al., 2015).” The authors should explain in more detail how this fact impacts the discussion of calcium isotope variability in Nautilus shells.

Line 285: “The δ13C metabolic rate proxy is more advanced in fish but holds promise for cephalopods” Here I think the authors should give a bit more background about this proxy. Most notably how it was developed and what the underlying assumptions are.

Line 296: “bulk samples” If I understood the sampling protocol correctly, these are samples from the aperture. If I’m not mistaken, this should mean that they represent the part of the shell that mineralized last. If more detailed information is known about the life cycle of Nautilus and/or their living depth at the time of depth, the authors could produce a more reliable estimate of the living depth during deposition of the sampled shell material.

Line 300: “(~15%)” The reference to increase of the fraction of respired carbon in the shell as a percentage is slightly confusing, since this could be misread as a 15% increase relative to the original value (not an increase in absolute percentage). Perhaps this can be clarified.

Line 308: Many mollusks show marked changes in d13C of their shells over their lifetimes (see e.g. McConnaughey and Gillikin, 1997 and references therein). Can such an ontogenetic trend be excluded as an explanation for the observed difference between the samples? If so, the authors should motivate why this explanation is not considered here in the discussion.

Line 327: I am not convinced that there is more work done on diagenetic assessment in aragonites than in calcites. There exist many studies that focus on calcite alteration (e.g. Brand and Veizer, 1980; Al-Aasm and Veizer, 1986a; b; Ullmann and Korte, 2015). Perhaps this statement can be revised or clarified to better reflect the literature.

Figures 4-9: In all these figures, I would ask the authors to more clearly highlight the uncertainty on the measurements. All symbols should be equipped with error bars in horizontal and vertical direction, and ideally these error bars should reveal the same metric (preferably 2 standard deviations of reproducibility or a 95% confidence level). Even if the error bars fall within the symbol, I would still think it improves the clarity of the data if the authors would make the symbol smaller and/or represent the data points as crossover points of error bars. This way, it is more straightforward for the reader to judge whether the differences between datapoints and between groups of datapoints discussed in the text are statistically sound. In addition, if possible, it would be even better to include error ellipses (or combined error bars) of groups of datapoints that are discussed together (e.g. all sub-fossil samples or all modern specimens) on groups of specimens/measurements so the differences between groups is easier to interpret.

References

Al-Aasm I. S. and Veizer J. (1986a) Diagenetic stabilization of aragonite and low-Mg calcite, II. Stable isotopes in rudists. Journal of Sedimentary Research 56, 763–770.

Al-Aasm I. S. and Veizer J. (1986b) Diagenetic stabilization of aragonite and low-Mg calcite, II. Stable isotopes in rudists. Journal of Sedimentary Research 56.

Brand U. and Veizer J. (1980) Chemical diagenesis of a multicomponent carbonate system–1: Trace elements. Journal of Sedimentary Research 50, 1219–1236.

Gillikin D. P., Lorrain A., Jolivet A., Kelemen Z., Chauvaud L. and Bouillon S. (2017) High-resolution nitrogen stable isotope sclerochronology of bivalve shell carbonate-bound organics. Geochimica et Cosmochimica Acta 200, 55–66.

McConnaughey T. A., Burdett J., Whelan J. F. and Paull C. K. (1997) Carbon isotopes in biological carbonates: respiration and photosynthesis. Geochimica et Cosmochimica Acta 61, 611–622.Ren H., Sigman D. M., Meckler A. N., Plessen B., Robinson R. S., Rosenthal Y. and Haug G. H. (2009) Foraminiferal isotope evidence of reduced nitrogen fixation in the ice age Atlantic Ocean. Science 323, 244–248.

Straub M., Sigman D. M., Ren H., Martínez-García A., Meckler A. N., Hain M. P. and Haug G. H. (2013) Changes in North Atlantic nitrogen fixation controlled by ocean circulation. Nature 501, 200–203.

Ullmann C. V. and Korte C. (2015) Diagenetic alteration in low-Mg calcite from macrofossils: a review. Geological Quarterly 59, 3–20.

de Winter N. J., Dämmer L. K., Falkenroth M., Reichart G.-J., Moretti S., Martínez-García A., Höche N., Schöne B. R., Rodiouchkina K., Goderis S., Vanhaecke F., van Leeuwen S. M. and Ziegler M. (2021) Multi-isotopic and trace element evidence against different formation pathways for oyster microstructures. Geochimica et Cosmochimica Acta 308, 326–352.

Reviewer #2: This paper lays out a clear rationale for using historical samples to assess the meaning of geochemical signals in Nautilus shells. The manuscript is very well organized, and the analytical methods used are clearly documented. However, it is difficult to gather what the meaningful lessons from this dataset are. The stated goal is to ‘calibrate the geochemical response of cephalopod [analyses] to modern anthropogenic environmental change’, and yet it seems the historical environmental records of these variables and the ways that the proxies respond to such forcings are both too poorly defined to achieve this goal. What are the actual hypotheses that are being tested?

For example, there seems to be too many possibilities for forcing environmental d13C to know how to interpret the Nautilus signal. If the d13C signal is the result of anthropogenic changes, then why does the cenote sample fall between the historical anthropogenically-driven trend? Similarly, how can we interpret the substantial range of organic matter d13C that apparently cannot be attributed to diagenesis or diet or circulation (section 4.1.1)? There is discussion of 4-5 possible controls on d44/40Ca, but none is apparently important despite recent environmental changes because there is no change across most samples.

There is a lack of quantification in some areas of the discussion. The depth range of the Nautilus habitat should be more precisely described (line 295 says average depth of 400 m – this should come earlier and include variation and range). Line 177: ‘much lower d15N’ – how much lower? Comparison of Cretaceous mollusks (line 194): how much higher Sr/Ca and lower d44/40Ca in this other dataset, to aid comparison? To what degree could the cenote water composition vary from the open ocean? What is the magnitude of the anthropogenic Suess effect in d13C, against which the shell d13C is compared? What drives the lack of temporal change in Mg/Ca or Sr/Ca (these data are not discussed at all)?

The age axis on figures needs to be more accurate to include the 6800 year old age of the cenote sample, as well as the timing of indigenous settlement at 3000 BCE.

In summary, this paper needs to outline its hypotheses more clearly, and more clearly and quantitatively describe what the likely known or unknown forcings are, tested against the known or unknown effects of the proxies.

Reviewer #3: Linzmeier et al compared isotope data from modern, historical and subfossil Nautilus shells. A very useful set of data, worth to be published in PLoS One after some moderate changes.

The Introduction needs to be more specific on the goals of the study; testable hypotheses and research questions need to be provided. In the Discussion, these questions need to be picked up and answered.

Authors need to provide more analytical details (eg L114, L128-131 etc.).

The NBS-18 ref.mat is far away from the isotope values of the measured shells. It requires a two-point calibration with another, much higher standard, or more simply, you should have used a reference material closer to the samples, i.e., IAEA 603. I guess it is now too late to do that. Can you provide results from other certified materials (blindly measured ref.mats for quality control) or material with known isotope composition that have likewise been calibrated against NBS-18 in the past (or in the meantime) documenting the the accuracy of the data?

What do you measn with "precision", internal precision or external precision (= accuracy)?

I am not happy with the style of the figures. Symbols are too big, color gradient of scale is superfluous, x- and y-axis must be black lines, grey background and white lines need to be removed. Figure captions need to provide details on meaning of error bars.

Further comments are provided in the annotated pdf.

6. PLOS authors have the option to publish the peer review history of their article (what does this mean?). If published, this will include your full peer review and any attached files.

Reviewer #1: No

Reviewer #2: No

Reviewer #3: No

---

## [Author Response · Author response to Decision Letter 0]

12 Jul 2022

Dear Dr. Pucéat and reviewers,

We have addressed comments in both the cover letter and the response to reviewer document that have been uploaded. All authors wish to thank you for your time and expertise in evaluating this manuscript. We apologize for the delay in submission of revisions, but look forward to your feedback on this revised version of the manuscript.

Sincerely,

Benjamin Linzmeier

---

## [Decision Letter · Decision Letter 1]

13 Sep 2022

PONE-D-21-31441R1Isotope systematics of subfossil, historical, and modern Nautilus macromphalus from New CaledoniaPLOS ONE

Dear Dr. Linzmeier,

Thank you for submitting your manuscript to PLOS ONE. All reviewers have read your revised manuscript and recommend publication pending some minor editorial changes. I agree with the reviewers and have taken the liberty to make some additional minor suggestions for improvements (as comments in the attached pdf). Please carefully consider these suggestions and submit a revised manuscript by Oct 28 2022 11:59PM. The nature of the revision in my opinion does not require another round of review and is meant purely to improve the manuscript. If you will need more time than this to complete your revisions, please reply to this message or contact the journal office at plosone@plos.org. Please include the following items when submitting your revised manuscript:A rebuttal letter that responds to each point raised by the academic editor and reviewer(s). You should upload this letter as a separate file labeled 'Response to Reviewers'.A marked-up copy of your manuscript that highlights changes made to the original version. You should upload this as a separate file labeled 'Revised Manuscript with Track Changes'.An unmarked version of your revised paper without tracked changes. You should upload this as a separate file labeled 'Manuscript'.If applicable, we recommend that you deposit your laboratory protocols in protocols.io to enhance the reproducibility of your results. Protocols.io assigns your protocol its own identifier (DOI) so that it can be cited independently in the future. For instructions see: https://journals.plos.org/plosone/s/submission-guidelines#loc-laboratory-protocols. Additionally, PLOS ONE offers an option for publishing peer-reviewed Lab Protocol articles, which describe protocols hosted on protocols.io. Read more information on sharing protocols at https://plos.org/protocols?utm_medium=editorial-email&utm_source=authorletters&utm_campaign=protocols.

We look forward to receiving your revised manuscript.

Kind regards,

Lukas Jonkers

Academic Editor

PLOS ONE

Journal Requirements:

Reviewers' comments:

Reviewer's Responses to Questions

**Comments to the Author**

1. If the authors have adequately addressed your comments raised in a previous round of review and you feel that this manuscript is now acceptable for publication, you may indicate that here to bypass the “Comments to the Author” section, enter your conflict of interest statement in the “Confidential to Editor” section, and submit your "Accept" recommendation.

Reviewer #1: All comments have been addressed

Reviewer #2: All comments have been addressed

Reviewer #3: (No Response)

2. Is the manuscript technically sound, and do the data support the conclusions?

Reviewer #1: Yes

Reviewer #2: Yes

Reviewer #3: Yes

3. Has the statistical analysis been performed appropriately and rigorously? 

Reviewer #1: Yes

Reviewer #2: Yes

Reviewer #3: Yes

4. Have the authors made all data underlying the findings in their manuscript fully available?

Reviewer #1: Yes

Reviewer #2: Yes

Reviewer #3: Yes

5. Is the manuscript presented in an intelligible fashion and written in standard English?

Reviewer #1: Yes

Reviewer #2: Yes

Reviewer #3: Yes

6. Review Comments to the Author

Reviewer #1: Dear Authors,

Thank you for your comprehensive reply to the review comments, and for submitting a thoroughly revised manuscript. In my opinion, all the queries I have posed in my review have been addressed and I appreciate the way the authors responded to the other review comments as well. The manuscript has improved significantly since the previous version and I am happy to support publication in its current form.

Reviewer #2: The authors have completed a solid revision of their manuscript that presents their results in helpful and appropriate context, including possible mechanisms explaining the variation (and ruling out some that do not) and guiding the reader to remaining questions and room for further work. I think this contribution will be valuable to others seeking to perform robust calibrations of skeletal-based geochemical proxies.

Reviewer #3: Most of the comments raised in my first review have been properly addressed by Benjamin Linzmeier and colleagues.

However, the authors need to follow common rules in abbreviating taxonomic names. I did not ask to abbreviate the species name (macromphalus), but the genus name, Nautilus, after the first mention in the ms. Hence, Nautilus macromphalus becomes N. macromphalus with the second and following occurences in the main text. Genus and species names should still spelled out in the figure captions and not abbreviated.

It still falsely reads "Carrera"! The Italian city is spelled Carrara (with three "a"s), and so is the corresponding marble. The company Carrera produces racetracks for kids.

I still do not understand the relevance of Figure 3C. If the authors want to show how the microstructure of the different shell layers looks like, nacreous and prismatic, they need to provide a high-resolution close-up (SEM) image that reveals the individual nacre tablets and prisms. The current figure does not provide any useful information.

7. PLOS authors have the option to publish the peer review history of their article (what does this mean?). If published, this will include your full peer review and any attached files.

Reviewer #1: No

Reviewer #2: No

Reviewer #3: No

---

## [Author Response · Author response to Decision Letter 1]

27 Oct 2022

Editor Line-by-line comments:

Line 59: y

Changed

Line 64: perhaps cite the earliest study/ies

In [20], some unpublished data on Nautilus are presented and discussed. The reference itself is a book chapter. Since initial submission of this paper, others [21] have published a few additional δ44/40Ca analyses of Nautilus and other cephalopods.

Line 84: perhaps mention earlier that the samples are from collections

We have added a clause in the first sentence.

Line 94: in italics and N. macromphalus, as reviewer suggested

We have addressed this through the manuscript.

Line 97: red? What are the abbreviations?

Changed and notes on the abbreviations (museum codes) added.

Line 118: are these radiocarbon or calendar years?

These are radiocarbon ages reported as years before present.

Line 181: perhaps write in full: amperes, or use the standard abbreviation A

This has been changed.

Line 189: is this how the data are made publicly available? Why not make them even FAIRer and deposit them in a known data repository (pangaea.de seems a good choice for this type of work; https://journals.plos.org/plosone/s/recommended-repositories)

We will deposit them in Earthchem to be FAIRer with these data. Pangaea.de has a fee for curation of data and takes a bit longer (6 weeks vs 2 weeks) to curate.

Line 281: s

Added.

Line 387: the formula (and lack of unit) suggest this is a fraction or proportion, rather than quantity?

Changed to proportion.

Line 386: adaptation?

We are not stating strict adaptation because there is not selection but rather some sort of phenotypic response that may take adjustment during development to express. From Fregley, 1996): Acclimation is a phenotypic response developed by an animal to a environmental stressor.

Fregley, M. J . 1996. Adaptations: some general characteristics. In: Fregley, M. J., and C. M.Blatteis, editors. Handbook of physiology, section 4: environmental physiology. vol. I. Oxford: Oxford University Press; p. 3–15.

Reviewer #1: Dear Authors,

Thank you for your comprehensive reply to the review comments, and for submitting a thoroughly revised manuscript. In my opinion, all the queries I have posed in my review have been addressed and I appreciate the way the authors responded to the other review comments as well. The manuscript has improved significantly since the previous version and I am happy to support publication in its current form.

 Thank you for your work in critiquing the first and revised versions of the manuscript.

Reviewer #2: The authors have completed a solid revision of their manuscript that presents their results in helpful and appropriate context, including possible mechanisms explaining the variation (and ruling out some that do not) and guiding the reader to remaining questions and room for further work. I think this contribution will be valuable to others seeking to perform robust calibrations of skeletal-based geochemical proxies.

 Thank you for both your first review and your work to check through the revision.

Reviewer #3: Most of the comments raised in my first review have been properly addressed by Benjamin Linzmeier and colleagues.

However, the authors need to follow common rules in abbreviating taxonomic names. I did not ask to abbreviate the species name (macromphalus), but the genus name, Nautilus, after the first mention in the ms. Hence, Nautilus macromphalus becomes N. macromphalus with the second and following occurences in the main text. Genus and species names should still spelled out in the figure captions and not abbreviated.

Thank you for catching this. I’m not entirely sure why I ended up doing that style of abbreviation, but it has been changed.

It still falsely reads "Carrera"! The Italian city is spelled Carrara (with three "a"s), and so is the corresponding marble. The company Carrera produces racetracks for kids.

We are using the Carrara Marble from Italy and not the marble from General Carrera Lake in Chile/Argentina. It has been changed to the proper spelling.

I still do not understand the relevance of Figure 3C. If the authors want to show how the microstructure of the different shell layers looks like, nacreous and prismatic, they need to provide a high-resolution close-up (SEM) image that reveals the individual nacre tablets and prisms. The current figure does not provide any useful information.

 We have removed it.

---

## [Editor Report · Decision Letter 2]

2 Nov 2022

Isotope systematics of subfossil, historical, and modern Nautilus macromphalus from New Caledonia

PONE-D-21-31441R2

Dear Dr. Linzmeier,

We’re pleased to inform you that your manuscript has been judged scientifically suitable for publication and will be formally accepted for publication once it meets all outstanding technical requirements.

Kind regards,

Lukas Jonkers

Academic Editor

PLOS ONE

Additional Editor Comments (optional):

Thank you for submitting your data to an open repository. Please do not forget to include a link to the data in the manuscript.

---

## [Editor Report · Acceptance letter]

29 Nov 2022

PONE-D-21-31441R2 

Isotope systematics of subfossil, historical, and modern *Nautilus macromphalus* from New Caledonia 

Dear Dr. Linzmeier:

I'm pleased to inform you that your manuscript has been deemed suitable for publication in PLOS ONE. Congratulations! Your manuscript is now with our production department. 

Kind regards, 

on behalf of

Dr. Lukas Jonkers 

Academic Editor

PLOS ONE